# Obstacles against the Marketing of Curcumin as a Drug

**DOI:** 10.3390/ijms21186619

**Published:** 2020-09-10

**Authors:** Kambiz Hassanzadeh, Lucia Buccarello, Jessica Dragotto, Asadollah Mohammadi, Massimo Corbo, Marco Feligioni

**Affiliations:** 1European Brain Research Institute (EBRI) Rita Levi Montalcini Foundation, Viale Regina Elena 295, 00161 Rome, Italy; k.hassanzadeh@ebri.it (K.H.); l.buccarello@ebri.it (L.B.); j.dragotto@ebri.it (J.D.); 2Department of Biotechnology and Applied Clinical Sciences, University of L’Aquila, 67100 L’Aquila, Italy; 3Cellular and Molecular Research Center, Research Institute for Health Development, Kurdistan University of Medical Sciences, Sanandaj 66177-15175, Iran; Asadollah.Mohammadi@muk.ac.ir; 4Department of Neurorehabilitation Sciences, Casa Cura Policlinico, 20144 Milano, Italy; m.corbo@ccppdezza.it

**Keywords:** curcumin, pharmacokinetic, bioavailability, marketing

## Abstract

Among the extensive public and scientific interest in the use of phytochemicals to prevent or treat human diseases in recent years, natural compounds have been highly investigated to elucidate their therapeutic effect on chronic human diseases including cancer, cardiovascular disease, and neurodegenerative disease. Curcumin, an active principle of the perennial herb *Curcuma longa*, has attracted an increasing research interest over the last half-century due to its diversity of molecular targets, including transcription factors, enzymes, protein kinases, growth factors, inflammatory cytokines, receptors, and it’s interesting pharmacological activities. Despite that, the clinical effectiveness of the native curcumin is weak, owing to its low bioavailability and rapid metabolism. Preclinical data obtained from animal models and phase I clinical studies done in human volunteers confirmed a small amount of intestinal absorption, hepatic first pass effect, and some degree of intestinal metabolism, might explain its poor systemic availability when it is given via the oral route. During the last decade, researchers have attempted with new pharmaceutical methods such as nanoparticles, liposomes, micelles, solid dispersions, emulsions, and microspheres to improve the bioavailability of curcumin. As a result, a significant number of bioavailable curcumin-based formulations were introduced with a varying range of enhanced bioavailability. This manuscript critically reviews the available scientific evidence on the basic and clinical effects and molecular targets of curcumin. We also discuss its pharmacokinetic and problems for marketing curcumin as a drug.

## 1. Introduction and History

Natural ingredients have been used during human history for numerous purposes, including alimentary usages but, due to their pharmacological effects, it has grown in the years the scientific interest to use these compounds to prevent or treat human diseases. Therefore, recent years have witnessed a growing understanding of the effect of natural products as sources of new supplements and drugs. Curcumin is one of such compounds with a history that goes back to 5000 years before [1]. Turmeric derived from the rhizome of the plant *Curcuma longa* (*C. longa*) has been used by the Indian people for centuries with no known side effects, not only as a food color and flavor but also to treat a wide variety of illnesses [2]. Turmeric was stated in the writings of Marco Polo regarding his journey to India and China in 1280 AC and it was introduced to Europe, for the first time, in the 13th century by Arab traders [1].

The main polyphenol constituents of Turmeric (*C. longa*) are curcuminoids that have three main chemical components, including curcumin (75–80%), which was found to be the key colored compound, demethoxycurcumin (15–20%), and bisdemethoxycurcumin (3–5%) [3,4]. Curcumin is the yellow pigment of the Indian Spice Turmeric and it was first isolated almost two centuries ago in 1815 by two German scientists, Vogel and Pelletier. Curcumin is a water-insoluble powder obtained in crystalline form in 1870 [5] and eventually identified as 1,6-heptadiene-3,5-dione-1,7-bis(4-hydroxy-3-methoxyphenyl)-(1E,6E)or diferuloylmethane [6].

According to the PubMed database, the first report on antibacterial effect of curcumin was published in 1949 in Nature and the first clinical trial was published in The Lancet in 1937. Although PubMed indicates more than 9200 publications on curcumin, interestingly almost 7500 of those have been published in the last 10 years.

Scientific documents are still published, focusing on the different activity and therapeutic efficacy of curcumin. In 2015, Kumar and colleagues [7] created and developed the Curcumin Resource Database (CRDB) that aims to perform as a gateway to access all relevant data and related information on curcumin and its analogs. This database involves 1186 curcumin analogs, 195 molecular targets, 9075 peer-reviewed publications, 489 patents, and 176 varieties of *C. longa* obtained by extensive data mining and careful curation from numerous sources [7]. Curcumin has been commonly used in traditional Indian medicine to treat a wide variety of illnesses including hepatic disorders, rheumatism, biliary disorders, anorexia, cough, and sinusitis. Turmeric paste is also available in many Indian homes for treatment of inflammation and wounds.

In recent years, a huge amount of reports and information regarding the different roles of curcumin has been released. Curcumin has been found as a potent antioxidant and reactive oxygen/nitrogen species (ROS, RNS) scavenger [8]. Besides, it has been reported to induce an upregulation of antioxidant enzymes such as superoxide dismutase, glutathione peroxidase and reductase, catalase, etc. [9].

The other main function of curcumin goes back to its protective activity on mitochondrial function and dynamic [10,11] and prevention of neuroinflammation [12], and apoptosis [13,14].

Concerning the above background and history of curcumin, in this paper, we review several aspects including cellular studies to clinical trials related to curcumin, its cellular and molecular targets, its cellular toxicity, and the problems related to its low pharmacokinetic and the current attempts to increase curcumin absorption and bioavailability.

Finally, we discuss the fact that, despite a wide range of cellular, molecular, and clinical studies and interesting results, why curcumin has not yet been licensed as a drug and it couldn’t play a higher-order role in treatment of diseases?

## 2. Disease Targets of Curcumin: From Cell Lines to Clinical Trials Studies

In this section, we will discuss the several effects of curcumin, including antimicrobial, gastrointestinal, cardiovascular, anti-cancer, ant-inflammatory, neuroprotective in different disorders which have been previously reported. These findings, comprising cell line studies, animal studies, and clinical trials. Then, the possible mechanisms will be described based on disease categories. As shown in Figure 1, there is a wide variety of studies on curcumin ranging from gastrointestinal, central nervous system, and cardiovascular effects, to antimicrobial and anti-cancer characteristics.

### 2.1. Antimicrobial Effect of Curcumin

As shown in Table 1, there are some in vitro studies with different kinds of curcumin extracts and few number of clinical studies indicating the antimicrobial property of curcumin against organisms like bacteria, fungi, parasite and virus. Antimicrobial studies started with anti-parasitic effect of curcumin and the anti-bacterial characteristic has attracted the most attention.

The stability of FtsZ protofilaments as a vital factor for bacterial cytokinesis has been introduced as a drug target for the antibacterial agents. Inhibition of assembly dynamics of FtsZ in the Z-ring may suppress the bacterial cell proliferation as one of the probable curcumin antibacterial mechanisms of action [15]. Accumulated evidence indicated curcumin plays an inhibitory role against numerous viral infections. The mechanisms involved in antiviral effect of curcumin are either a direct interference of viral replication machinery or suppression of cellular signaling pathways essential for viral replication, such as phosphoinositide 3-kinase (PI3K)/protein kinase B (AKT), nuclear factor kappa-light-chain-enhancer of activated B cells (NF-κB) [16].

Studies indicating antimicrobial effects of curcumin in the last decade focused on the pharmacokinetics of curcumin rather than pharmacodynamics, reflecting the concerns of researchers in this area. There is evidence suggesting that mode of antimicrobial activity of curcumin depends on the properties of the delivery system [17]. For instance, in an attempt to compare the antimicrobial activity toward *Escherichia coli* of two curcumin formulations: methyl-β-cyclodextrin supramolecular inclusion complex and polyelectrolyte-coated monolithic nanoparticles, researchers showed that while curcumin–β-cyclodextrin complexes exhibited a potent bactericidal activity, the curcumin nanoparticles were bacteriostatic [17].

**Table 1 ijms-21-06619-t001:** Antimicrobial Effect of Curcumin.

Product	Dose or Concentration Used	Effect and Findings	Type of Study	Studied by
Curcuminoids	Concentration 0.1–1 mg/mL	Anti-parasitic Nematocidal activity of mixed Curcuminoids	In vitro	Kiuchi F et al. 1993 [18]
Curcumin Curcumin + boric acid + oxalic acid dihydrate (boron complex)	Curcumin: IC50: 100 µM boron complex: IC50: 6 µM	Anti-viral inhibitor of the HIV-l	In vitro	Sui Z et al. 1993 [19]
Curcumin essential oil extracted	Concentration of 0.1% in the medium	Anti-fungal aflatoxin synthesis by *Aspergillus parasiticus*	In vitro	Tantaoui-Elaraki A et al. 1994 [20]
Curcumin	2.5 g which was repeated 7 days later	Anti-viral inhibitor of the HIV-1	Clinical Trial (3 Subjects)	Jordan W.C et al. 1996 [21]
Turmeric extract	antifungal activity against *Candida albicans* at 1 µg/mL	Anti-fungal *Candida albicans*	In vitro	Roth GN et al. 1998 [22]
Turmeric oil	Anti-bacterial activity in 50–200 ppm	Anti-bacterial	In vitro	Negi P.S et al.1999 [23]
Curcumin	cytotoxicity against leishmania in vitro. The LD50 = 37.6 µM	Anti-parasitic against leishmania	In vitro	Koide T et al. 2002 [24]
Curcumin extract	Anti-fungal at 50–500 mg/L	Anti-fungal	In vitro	Kim MK et al. 2003 [25]
Curcumin	In vitro: IC50: 5 µM Animal: once daily for 5 days at a dose of 100 mg/kg	Anti-malarial	In vitro & Animal model	Reddy RC et al. 2005 [26]
Curcumin	at 30 and 100 μM	Anti-parasitic *Giardia lamblia trophozoites*	In vitro	Pérez-Arriaga L et al. 2006 [27]
Curcumin	30 mg every 12 h for 7 days	Anti-bacterial	Clinical trial (25 Subjects)	Di Mario F et al. 2007 [28]
Curcumin extract quercitin and curcumin (FlogMEV) extracts	In patients with prostatitis quercitin (100 mg) and curcumin (200 mg) for 14 days	Anti-bacterial	Clinical trial (284 Subjects)	Cai T et al. 2009 [29]
Curcumin nanoparticle	Concentration of 260 μM	Anti-bacterial	In/vitro	Trigo Gutierrez JK et al. 2017 [30]
Curcumin nanoparticle	0.1 and 0.2 mg per well concentration	Anti-bacterial	In vitro	Fakhrullina G et al. 2019 [31]
Curcumin nanoparticle curcumin-silver nanoparticles	Minimum inhibitory concentration 20 mg/L	Anti-bacterial	In vitro	Jaiswal S et al. 2018 [32]
Iodinated curcumin	Minimum inhibitory concentration 150 and 120 µg/mL	Anti-bacterial	In vitro	Manchanda G et al. 2018 [33]

In Table 1, there is an overview of the multipotent antimicrobial character of curcumin that would be useful for investigators to additional studies on curcumin’s antimicrobial role against new microorganisms.

### 2.2. Gastrointestinal Effect of Curcumin

Similar to the antimicrobial studies, the gastrointestinal activity of curcumin has been tested in animal models and there are also few human clinical studies. Curcumin has been suggested as a remedy for liver and digestive diseases such as irritable bowel syndrome, colitis, Crohn’s disease and bacterial and parasitic diseases. Data in Table 2 shows that most of the reports are related to liver protection and some limited studies indicate the intestine and colon protection properties of curcumin. On the other hand, some reports are about its beneficial effects in animal models recapitulating inflammatory bowel disease. In addition to these effects, in several animal studies, curcumin has been revealed to influence the composition of gut microbiome. For instance, adding the curcumin to a high-fat diet in animals, prevented the fat-induced changes of the gut microbiota [34].

Several lines of evidence support the hepatoprotective effect of curcumin. The findings in this area indicated that curcumin exerts notable protective and therapeutic effects in oxidative liver diseases through numerous molecular mechanisms including the pro-inflammatory cytokines and lipid perodixation products suppression, activation of PI3K/AKT and hepatic stellate cells and ameliorating cellular responses to oxidative stress [35]. As shown in Table 2, the data is mostly obtained from animal studies and further clinical studies are needed to clarify the mechanisms of curcumin in oxidative associated liver diseases.

Research on colitis in animal models indicate a significant decrease in pro-inflammatory biomarkers and modulation of inflammatory cells [36,37,38]. In a study, it has been strongly suggested that curcumin diminishes neutrophil recruitment to the inflammatory sites through interference in chemokine gradient formation in addition to the direct effect on neutrophil polarization, chemotaxis, and chemokinesis [38]. These mechanisms likely significantly contribute to the described protective and potent effect of curcumin in ulcerative colitis. In patients with ulcerative colitis, curcumin has been well-tolerated and led to ameliorate the symptoms and inflammatory markers in clinical trials [39,40].

Curcumin has been reported to be promising in the treatment of intestinal inflammatory diseases (IBD). It provides some beneficial effects on the microbiome, inhibition of toll-like receptors (TLR4)/NF-κB/activator protein1 (AP-1) signal transduction, alterations in cytokine profiles and immune cell maturation and differentiation. These are possible mechanisms contributing to IBD pathology thatare affected by curcumin to strengthen the intestinal barrier [41].

Effects of curcumin on the microbiome have been widely studied and it has been reported that these effects are different based on the disease characteristics. For instance, in a research on mice living in specific-pathogen-free conditions, curcumin reduced the microbial richness and diversity [42]. In another study on rat model of hepatic steatosis, administration of curcumin decreased species richness and diversity induced notable microbiota compositional changes compared to both high-fat diet and control groups. Also, it shifted the structure of the gut microbiota [43]. These microbiome changes inhibit intestinal inflammation. In fact, curcumin maintains short-chain fatty acid-producing bacteria, which acts to provide intestinal mucosal protection [44,45]. Table 2 shows some studies indicating the effect of curcuminoids and curcumin extract on different pathological conditions in the gastrointestinal system ranging from cell line to clinical studies.

**Table 2 ijms-21-06619-t002:** Gastrointestinal Effect of Curcumin.

Product	Dose or Concentration Used	Effect and Findings	Type of Study	Studied by
Curcumin	Concentrations 5–30μM	Liver protective through inhibiting hepatic stellate cells activation	In vitro	Tang Y et al. 2010 [46]
Curcumin	Dose: 25 μg daily for 10 weeks, intraperitoneal	Liver protective: effectively limits the development and progression of fibrosis	Animal model	Vizzutti F et al. 2010 [47]
Curcumin	300 mg/kg, by gavage daily for 12 weeks	Liver protective: inhibited the development of liver cirrhosis mainly due to its anti-inflammatory activities and not by a direct anti-fibrotic effect	Animal model	Bruck R et al. 2007 [48]
Curcumin	1 g after the evening meal for 6 months	Ameliorate ulcerative colitis remission in patients with ulcerative colitis	Clinical trial (89 Subjects)	Hanai H et al. 2006 [39]
Curcumin	550 mg of curcumin twice daily for 1 month and then 550 mg three times daily for another month.	Reductions in concomitant medications Crohn’s disease	Clinical trial (5 Subjects)	Holt PRet al. 2005 [49]
Curcumin	Concentrations 10–30 μM	Ameliorate Inflammatory bowel disease: dose-dependent suppression of metalloproteinase-3 in colonic myofibroblasts from children and adults with active IBD	In vitro	Epstein J et al. 2010 [50]
Curcumin	Dose: 75 mg/kg/day orally daily for 6 weeks	Liver protective: prevents chronic alcohol-induced liver disease involving decreasing ROS generation and enhancing antioxidative capacity	Animal model	Rong S et al. 2012 [51]
Curcumin	Dose: 150 mg/kg, orally daily for 6 weeks	Liver protective: by inhibition of oxidative stress via mitogen-activated protein kinase/nuclear factor E2-related factor 2	Animal model	Xiong ZE et al. 2015 [52]
Curcumin	Dose: 150 mg/kg, orally daily for 8 weeks	Liver protective: prevention of the oxidative stress induced by chronic alcohol	Animal model	Varatharajalu R et al. 2016 [53]
Curcumin	Dose: 70 mg/kg, orally daily for 8 weeks	Liver protective: improvement of different features of Non-alcoholic fatty liver disease after a short-term supplementation	Clinical trial (80 Subjects)	Rahmani S et al. 2016 [54]
Curcumin	Curcumin (2%) diet from 4 to 18 weeks of age	Intestine protective: beneficial effects of dietary curcumin on intestinal tumorigenesis in rodent models of colon cancer.	Animal model	Murphy E.A et al. 2011 [55]
Curcumin	Dose: 50 mg/kg, orally daily for 10	Inflammatory bowel disease: beneficial effects in experimental colitis and may, therefore, be useful in the treatment of IBD.	Animal model	Ukil A et al. 2003 [56]
Curcumin	Curcumin (2%) diet from 9 days	ulcerative colitis: dietary curcumin may be of different value for Crohn’s disease and ulcerative colitis.	Animal model	Billerey-Larmonier C et al. 2008 [57]

### 2.3. Cardiovascular Protective Effect of Curcumin

The cardiovascular protective effects of curcumin have been extensively studied and its use as an adjuvant or therapeutic agent to alleviate cardiovascular disease and other vascular dysfunctions is currently being investigated. As shown in Table 3, the most evaluations on cardiovascular effects of curcumin have been made in the last decade and cardioprotective and anti-hyperlipidemia have more considered for curcumin. The majority of studies are in animal models using curcumin extract, while there are few assessments with nanoparticles.

Curcumin at a dose of 200 mg/kg, nine days (7 days before and 2 days following Adriamycin), significantly prevented adriamycin-induced cardiotoxicity. The mechanism of cardioprotective effects of curcumin is unclear and multiple mechanisms have been proposed in this area including (1) curcumin prevents lipid peroxidation through free radicals scavenging, leading to a blocking of the lipid chain reaction. (2) Curcumin exerts membrane-stabilizing effect, which is supported by the fact that it could prevent the ECG changes induced by adriamycin [58].

In diabetic patients with cardiovascular complications, curcumin has been shown to downregulate nitric oxide synthase (NOS) and reduce NO oxidation, which plays a key role in the pathogenesis of cardiovascular complications in diabetes [59]. In an animal study, researchers demonstrated that the myocardial tissue from diabetic rats exhibited increased levels of NOS enzyme mRNA as compared to control rats which was prevented by curcumin treatment showing a decrease in the oxidative DNA damage [60].

Cardiac hypertrophy is the heart’s response to some variety stimuli such as workload or myocardial infarction that increase biomechanical stress. It is characterized by an increase in the size of the individual cardiac myocytes and enlargement of the myocardium and the whole heart [61]. This disorder threatens affected patients with progression to overt heart failure or sudden death.

Histone acetylation is one of the important control points for the regulation of genes in the hypertrophic myocardium [62]. Curcumin has been reported to be an inhibitor of histone acetyltransferases (HATs), p300-HAT, which perform the acetylation of histone tails [63], therefore, it may have an effect in the prevention of the cardiac hypertrophy and heart failure [64].

There is also limited evidence on anti-hyperchlostrolemia effect of curcumin. In an animal study, administration of curcumin extract had hypolipidemic effects in high cholesterol-induced hypercholesterolemic mice. The authors reported that a mixture of *Nelumbo nucifera* leaf (NL) and *C. longa* provided more advanced protection against high cholesterol diet-related lipid accumulation and liver dysfunction and may be a more effective functional food for the management of hypercholesterolemia [65]. In another animal study, curcumin has been found to have hypocholesterolemic effect on both normal and hypercholesterolemic rats and was more effective in hypercholesterolemic animals. The possible proposed mechanism for this effect was, interfering with intestinal cholesterol uptake, augmentation of the conversion of cholesterol into bile acids, and increasing the excretion of bile acids [66].

Despite the extensive research regarding the effect of curcumin on the cardiovascular diseases in animals, additional studies in humans on curcumin’s cardiovascular role and mechanisms using new formulations exerting a high degree of bioavailability are required.

**Table 3 ijms-21-06619-t003:** Cardiovascular Protective Effect of Curcumin.

Product	Dose or Concentration Used	Effect and Findings	Type of Study	Studied by
Curcumin	Dose: 25–50–100–200 mg/kg, orally daily for 10 days	Cardioprotective: curcumin (50 mg/kg) with piperine (20 mg/kg) exhibited profound cardioprotection compared to curcumin (200 mg/kg) alone-treated group.	Animal model	Chakraborty M et al. 2017 [67]
Curcumin	Dose: 120 mg/kg, orally daily for 67 days	Cardioprotective: through direct antioxidant effects and indirect strategies that could be related to protein kinase C-activated downstream signaling.	Animal model	Correa F et al. 2013 [68]
Curcumin	Dose: 200 mg/kg, orally daily for 4 weeks	Cardioprotective: cardioprotective effect could be attributed to antioxidant.	Animal model	Swamy AV et al. 2012 [69]
Curcumin	Dose: curcumin (100 mg/kg) plus piperine (5 mg/kg) orally daily for 4 weeks	Anti-hypercholesterolemia: co-administration of curcumin plus piperine increasing the activity and gene expression of ApoAI, CYP7A1, LCAT, and LDLR, providing a promising combination for the treatment of hyperlipidemia.	Animal model	Tu Y et al. 2014 [70]
Curcumin	Dose: curcumin 100 mg/kg orally daily for 6 weeks	Cardioprotective: concomitant co-administration of curcumin and metformin revealed more protection than metformin alone through Inhibition of JAK/STAT pathway and activation of Nrf2/HO-1 pathway	Animal model	Abdelsamia E.M et al. 2019 [71]
Curcumin nanoparticle: curcumin and nisin based poly lactic acid nanoparticle (CurNisNp)	Dose: 10 and 21 mg/kg injection daily for 7 days	Cardioprotective: curcumin nanoparticle confers a significant level of cardioprotection in the guinea pig and is nontoxic.	Animal model	Nabofa W.E.E et al. 2018 [72]
Curcumin	Dose: curcumin 100 mg/kg orally daily for 24 days	Cardioprotective: Curcumin improve the heart function and structural changes in doxorubicin-treated rats	Animal model	Jafarinezhad Z et al. 2019 [73]
Curcumin nanoparticle	Dose: 100–150 mg/kg orally daily for 15 days	Cardioprotective: curcumin nanoparticles exert better antioxidative effects on MI compared to conventional curcumin, thus improving myocardial function more effectively and limiting the extension of heart damage.	Animal model	Boarescu PM, et al. 2019 [74]
Curcumin	Dose: 100 mg/kg orally daily for 7 days	Cardioprotective: protects against myocardial infarction-induced cardiac fibrosis via SIRT1 activation	In vitro and in vivo	Xiao J et al. 2016 [75]
Curcuminoids	Dose: 4 g orally daily for 8 days	Cardioprotective: significantly decreased MI associated with coronary artery bypass grafting through the antioxidant and anti-inflammatory effects	Clinical trial (121 Subjects)	Wongcharoen W et al. 2014 [76]
Curcumin	Concentration: 5 μmol/L	Vascular protective: effectively reverses the endothelial dysfunction induced by homocysteine	In vitro	Ramaswami G et al. 2004 [77]
Curcumin	Curcumin (0.05-g/100-g diet) for 10 weeks	Anti-hyperlipidemia: curcumin exhibits an obvious hypolipidemic effect by increasing plasma paraoxonase activity, ratios of high-density lipoprotein cholesterol to total cholesterol and of apo A-I to apo B, and hepatic fatty acid oxidation activity with simultaneous inhibition of hepatic fatty acid and cholesterol biosynthesis in high-fat–fed hamsters.	Animal model	Jang EM et al. 2008 [78]
Curcumin	Curcumin (0.02% *w*/*w* diet) for 18 weeks	Anti-atherogenic: Long-term curcumin treatment lowers plasma and hepatic cholesterol and suppresses early atherosclerotic lesions comparable to the protective effects of lovastatin.	Animal model	Shin S.K et al. 2011 [79]
Curcumin extract: hydro-alcoholic extract of rhyzome of *C. longa* containing ∼10 mg of curcumin	Dose: 20 mg orally daily for 30 days	Anti-hyperlipidemia: decreases significantly the LDL and apo B and increases the HDL and apo A of healthy subjects	Clinical trial (30 Subjects)	Ramírez-Boscá A et al. 2000 [80]

### 2.4. Anti-Cancer Effect of Curcumin

Numerous molecular targets have been proposed concerning the chemotherapeutical effect of curcumin against different types of cancer. Extensive studies, mainly in vitro or in animal models, indicate that it can modulate all kinds of cancer hallmarks, including uncontrolled cell proliferation, cancer-associated inflammation, cancer cell death, signaling pathways, cancer angiogenesis, and metastasis [81]. According to a report on curcumin in 2019, 37% of all researches on curcumin focus on its anti-cancer effect or relevant mechanisms. As shown in Table 4, effect of curcumin in different organ cancers have been widely studied, almost all studies have been done in vitro or in animal models. Based on these reports, curcumin, through different signaling pathways, represents a promising candidate as an effective anti-cancer agent to be used alone or in combination with other drugs. It affects molecular targets involved in the development of several cancers.

Several lines of evidence revealed that inflammatory pathways disorder plays a key role in cancer development [82]. Increase in the production of pro-inflammatory cytokines such as tumor necrosis factor alpha (TNF-α) and interleukins, transcription factors including nuclear factor κB (NF-κB), ROS, cyclooxygenase (COX-2), protein kinases B (AKT), activator protein 1 (AP1), signal transducer and activator of transcription 3 (STAT3), causing the initiation and development of cancer [83,84].

Curcumin exerts its immunomodulatory characteristics by interacting with above-mentioned immune mediators, hence its anti-cancer property.

There are few clinical trials for use of curcumin in cancers, either as a monotherapy or in combination with other anti-cancer agents. In a phase I clinical trial, curcumin was administered orally in 15 colorectal cancer patients. The findings reveal that there was not a significant toxicity except a diarrhea symptom reported in two patients, and two patients showed stable disease after two months of curcumin treatment [85]. Another monotherapy phase II clinical trial study of oral formulation of curcumin administered to 25 advanced pancreatic cancer patients showed low levels of curcumin in plasma (22–41 ng/mL) but two patients showed clinical biological activity. In one patient, it was reported disease stability for >18 months. In another patient, a brief but marked tumor regression (73%) was found [86].

There are some combination therapy studies using curcumin in combination with other chemotherapeutic agents used in standard treatments of cancer disease. In a clinical study, a combination therapy of curcumin with imatinib (tyrosine kinase inhibitor) has been evaluated in 50 chronic myeloid leukemia patients in which the synergic action of the two drugs was more efficient than imatinib alone, although additional studies are needed to confirm this efficacy [87]. Furthermore, a combination of curcumin with anti-EGFR (epidermal growth factor receptor) monoclonal antibodies in cutaneous squamous cell carcinoma patients has been shown as a highly effective strategy in disease control in another clinical [88]. Table 4 summarized some of studies indicating anti-cancer effects of curcumin in different organs in the last two decades.

**Table 4 ijms-21-06619-t004:** Anti-cancer Effect of Curcumin.

Product	Dose or Concentration Used	Effect and Findings	Type of Study	Studied by
Curcumin	Concentration 15 µM	Prostate cancer. chronic inflammation can induce a metastasis prone phenotype in prostate cancer cells: Curcumin disrupts this feedback loop by the inhibition of NFκB signaling	In vitro	Killian PH et al. 2012 [89]
Curcumin	Concentration 50 µM for 1–4 h	Colon cancer: curcumin is an activator of PTPN1 and can reduce cell motility in colon cancer via dephosphorylation of pTyr(421)-CTTN, which could be exploited for novel therapeutic approaches in colon cancer	In vitro	Radhakrishnan VM et al. 2014 [90]
Curcumin or tetrahydrocurcumin (THC)	Curcumin: 300 mg/kg THC: 3000 mg/kg for 21 days	Anti-cancer: anti-angiogenic properties of Curcumin and THC represent a common potential mechanism for their anti-cancer actions.	Animal model	Yoysungnoen P et al. 2008 [91]
Curcumin	Concentration 0–20 μM	Breast cancer: curcumin suppresses chemokine-like ECM-associated protein osteopontin-induced VEGF expression and tumor angiogenesis	In vitro	Chakraborty G et al. 2008 [92]
Curcumin	Concentration 3.12–50 µM	ovarian and endometrial cancers: curcumin suppresses JAK-STAT signaling via activation of PIAS-3, thus attenuating STAT-3 phosphorylation and tumor cell growth.	In vitro	Saydmohammed M et al. 2010 [93]
Curcumin	Concentration 20–40 µM	Liver cancer: suppresses migration and proliferation of Hep3B hepatocarcinoma cells through inhibition of the Wnt signaling pathway	In vitro	Kim HJ et al. 2013 [94]
Curcumin	Concentration (2, 20, and 50 μM) for 4 h	Burkitt’s lymphoma: curcumin might play an important role in radiotherapy of high-grade non-Hodgkin’s lymphoma through inhibition of the PI3K/AKT-dependent NF-κB pathway.	In vitro	Qiao Q et al. 2013 [95]
Curcumin	Concentration 0–20 μg/mL for 24 h	Osteosarcoma: curcumin caused death of HOS cells by blocking cells successively in G(1)/S and G(2)/M phases and activating the caspase-3 pathway	In vitro	Lee DS et al. 2009 [96]
Curcumin	Concentration 4–10 µM for 24 h	Glioma: curcumin exerts inhibitory action on glioma cell growth and proliferation through induction of cell cycle arrest	In vitro	Liu E et al. 2007 [97]
Curcumin	Concentration 10, 25 µM for 24 h	Breast cancer: Curcumin induces apoptosis in human breast cancer cells through p53-dependent Bax induction	In vitro	Choudhuri T et al. 2002 [98]
Curcumin	Concentration 0 to 20 μM for 24 h	Gastric carcinoma: curcumin inhibited the growth of the AGS cells and induced apoptosis	In vitro	Cao AL et al. 2015 [99]
Curcumin	Concentration 0 to 100 μM for 72 h	Adenocarcinoma: curcumin-induced growth inhibition through G2/M arrest in Ras-driven cells and by apoptosis induction in Src-driven cells,	In vitro	Ono M et al. 2013 [100]
Curcumin	Concentration 0 to 40 μM for 24–72 h	Colon cancer: Curcumin suppresses proliferation of colon cancer cells by targeting Cyclin-dependent kinase 2	In vitro	Lim TG et al. 2014 [101]
Curcumin micelles	Concentration 0 to 100 μg/mL for 24 h	Lung cancer: mixed micelles of PF127 and GL44 significant improvement in curcumin oral bioavailability.	In vitro	Patil S et al. 2015 [102]
Curcuminoids	Dose: 8 caplets daily for 8 weeks. Each caplet contains 1 g curcuminoids (900 mg curcumin, 80 mg desmethoxycurcumin, and 20 mg bisdesmethoxycurcumin)	Pancreatic cancer: Oral curcumin has biological activity in some patients with pancreatic cancer.	Clinical trial (25 cases)	Dhillon N et al. 2008 [86]
Curcumin	Dose: 0.45 and 3.6 g daily for up to 4 months.	Colorectal cancer: a daily dose of 3.6 g of curcumin are suitable for its evaluation in the prevention of malignancies at sites other than the gastrointestinal tract.	Clinical trial (15 cases) phase I	Sharma RA et al. 2004 [85]

### 2.5. Effect of Curcumin on Skin Diseases

Growing evidence suggests that curcumin may show an effective role in the treatment of several skin disorders [103] as shown in the provided in the table below. The studies imply on the anti-inflammatory effect of curcumin except those that show apoptotic effects of curcumin for cancer indication (Table 5). Study on the possible interactions between curcumin and other chemicals, commonly used in topical skin treatments, may provide useful insights for the development of new effective preparations, tailored for different conditions.

In the topical route of administration, curcumin showed a good efficiency, especially when incorporated in new formulations such as polymeric bandages, chitosan-alginate sponges, nano-emulsion, alginate foams, collagen films, hydrogel, and β-cyclodextrin-curcumin nanoparticle complex, making curcumin suitable as a therapeutic agent for the topical treatment of skin diseases [104,105,106,107,108,109,110].

**Table 5 ijms-21-06619-t005:** Therapeutic Effect of Curcumin in skin diseases.

Product	Dose or Concentration Used	Effect and Findings	Type of Study	Studied by
Curcumin	Dose: 40 mg/kg orally daily for 20 days	Psoriasis: all psoriasis indexes including ear redness, weight, thickness and lymph node weight were significantly improved	Animal model	Kang D et al. 2016 [111]
Turmeric tonic	Topical tonic Twice a day for 9 weeks	Psoriasis: turmeric tonic significantly reduced the erythema, scaling and induration of lesions (PASI score), and also improved the patients’ quality of life	Clinical trial (40 subjects)	Bahraini P et al. 2018 [112]
Curcumin nano-fiber chrysin-curcumin nano-fiber	Topical 5–7.5–10% *w*/*w* for 5, 10, 15 days	Wound healing: chrysin-curcumin-loaded nanofibers have anti-inflammatory properties in several stages of the wound-healing process by affecting the IL-6, MMP-2, TIMP-1, TIMP-2, and iNOS gene expression.	Animal model	Mohammadi Z et al. 2019 [113]
Curcumin nanocapsule	Dose: 6 mg/kg, intra-peritoneally, twice a week for 21 days	Skin cancer: curcumin caused significant reduction of cell viability in a concentration- and time-dependent manner.	Animal model	Mazzarino L et al. 2011 [114]
Curcumin	Concentration 0 to 20 μM for 6, 12 h	Skin cancer, melanoma: curcumin-induced cell death and apoptosis	In vitro	Yu T et al. 2010 [115]

### 2.6. Neuroprotective Effect of Curcumin

Curcumin has several appropriate characteristics for a neuroprotective agent, including potent antioxidant, anti-inflammatory, and anti-protein-aggregate activities that have been previously reviewed [116,117]. Due to its above-mentioned pluripotent benefits, curcumin has countless potential for the prevention of many neurological conditions that existing therapeutics are less than optimal. These disorders include Parkinson’s, Alzheimer’s, Huntingtin’s, brain injury, stroke, and aging [118]. Data in Table 6 show a summary of the main findings on the potential beneficial effects of curcumin against neurodegeneration, and principal attention has been given to Alzheimer’s and Parkinson’s disease (PD).

Alzheimer’s disease (AD) is the most prevalent form of age-related dementia and currently, there are millions of AD patients in the world and this number is expected to increase dramatically with the demographic shift toward a more aged population, unless preventive processes would be achieved [119]. Classically, AD is characterized by the accumulation of amyloid and tau aggregation with the development of neurodegeneration and cognitive defects.

Curcumin has been tested in animal model of Alzheimer’s disease in which not only reduced oxidative stress damage and inflammation markers, but it also mitigates amyloid plaques accumulation and cognitive deficits in rats [120,121]. The possible mechanism for curcumin protection against tau aggregates might be related to its antioxidant characteristics since the initial step for tau dimerization is driven also by oxidative damage, lipid peroxidation, or redox-regulated disulfides [118,122].

Evidence in recent years supports the efficacy of curcumin in PD. In both in vitro and in vivo models of PD curcumin could prevent oxidative stress toxicity by reducing the production of ROS and malondialdehyde and restoring GSH levels [123,124,125] which shields against alpha-synuclein-induced toxicity in the brain [123]. More specifically, antioxidative and anti-apoptotic activity of curcumin has been reported in in vitro studies, indicating the neuroprotective effect of curcumin on dopaminergic neurons [124,125].

More recently, we found that curcumin, at the concentration of 5 µM, protected neuroblastoma (SH-SY5Y) cells against H2O2-induced cell death by modulating of Small Ubiquitin-like Modifier (SUMO)-1-JNK(c-Jun N-terminal kinases)-TAU axis, indicating that curcumin might be a promising therapeutic agent against not only oxidative stress, but also pathologies characterized through SUMO, JNK, and Tau alterations in many neurodegenerative diseases [126].

Other mechanisms of the neuroprotective effect of curcumin are from oxidative damage by restoring mitochondrial membrane potential, upregulation of Cu-Zn superoxide dismutase, and inhibiting the production of intracellular ROS [127].

Although, in all these studies, authors are pleased with their findings, adapted from in vitro and animal models studies, more preclinical and clinical trials are needed to verify the exact effect of curcumin in neurodegenerative disease using different formulations.

**Table 6 ijms-21-06619-t006:** Neuroprotective Effect of Curcumin in Neurodegenerative Diseases.

Product	Dose or Concentration Used	Effect and Findings	Type of Study	Studied by
Curcumin a natural dietary supplement (NDS), containing extracts from *Curcuma longa*, silymarin, guggul, chlorogenic acid, and inulin	Dose: daily administration of NDS (0.9 mg/mouse) for 16 weeks	Neuroprotective: NDS exerts neuroprotective effects in high fat diet-fed mice by reducing brain fat accumulation, oxidative stress and inflammation, and improving brain insulin resistance.	Animal model	Nuzzo D et al. 2018 [128]
Curcuminoids	Concentration 0.1–30 μM few minutes before addition to artificial cerebrospinal fluid for the perfusion	Neuroprotective: curcuminoids can restore susceptibility for plastic changes in CA1 excitability that is injured by exposure to Aβ peptide and rescue sinking PS LTP in A β-peptide-exposed hippocampal CA1 neurons.	In vitro	Ahmed Tet al. 2011 [129]
Curcumin	Concentration 0–8 μM	Alzheimer’s Disease: curcumin effectively disaggregates Abeta as well as prevents fibril and oligomer formation	Animal model	Yang F et al. 2005 [130]
Curcuminoids	Concentration 10 μM	Alzheimer’s Disease: curcumin binds to Aβ oligomers and to Aβ fibrils	In vitro	Yanagisawa D et al. 2011 [131]
Curcumin	Concentration 0–30 μM	Alzheimer’s Disease: curcumin significantly attenuated β amyloid-induced radical oxygen species production and β-sheet structure formation.	In vitro	Shimmyo Y et al. 2008 [132]
Curcumin	Concentration 0–10 μM	Alzheimer’s Disease: curcumin downregulated the expression of amyloid precursor protein and amyloid-β in swAPP695-HEK293 cells, which was through miR-15b-5p	In vitro	Liu HY et al. 2019 [133]
Curcuminoids	Dose: 3–30 mg/kg	Alzheimer’s Disease: increased PSD-95, synaptophysin and camkIV expression levels in the hippocampus in the rat AD model	Animal model	Ahmed T et al. 2010 [134]
Ethanolic extract of turmeric	Dose: 80 mg/kg orally, daily for three weeks	Alzheimer’s Disease: effectively prevented cognitive deficits	Animal model	Ishrat T et al. 2009 [135]
Curcumin C3 Complex(^®^) an extract derived from the rhizomes (roots) of the plant *Curcuma longa*	Dose: 2, 4 g/day, orally for 24 weeks.	Alzheimer’s Disease: Results were unable to demonstrate clinical or biochemical evidence of efficacy of this formulation.	Clinical trial (36 Subjects)	Ringman JM et al. 2012 [136]
Tumeric powder capsules	Dose: 764 mg/day turmeric (100 mg/day curcumin) orally for 12 weeks	Alzheimer’s Disease: a significant improvement of the behavioral symptoms in the AD with the turmeric treatment,	Clinical trial (3 Subjects)	Hishikawa N et al. 2012 [137]
Curcumin	Concentration 0–1 μM	Parkinson’s Disease: Curcumin protected brain mitochondria against peroxynitrite by direct detoxification and inhibition of 3-nitrotyrosine formation and by elevation of total cellular glutathione levels in vivo	In vitro	Mythri RB et al. 2007 [138]
Curcumin nanoparticle polymeric nanoparticle encapsulated curcumin	In vitro: (1, 10, 50, 100, 500 nM, 1, 5 μM) In vivo: 25 mg/kg intraperitoneally twice daily for 4 weeks	Alzheimer’s Disease: NanoCurc™ ameliorated ROS-mediated damage in both cell culture and in animal models	Animal model/In vitro	Ray B et al. 2011 [139]
Curcumin	Concentration 0–10 μM for 24 h	Neuroprotection: curcumin enhanced neuronal survival against NMDA toxicity	In vitro	Lin MS et al. 2011 [140]
Curcumin	diet of 500 ppm curcumin for 4 weeks	Traumatic brain injury (TBI): curcumin reduced oxidative damage, normalized levels of BDNF, synapsin I, and CREB and counteracted the cognitive impairment caused by TBI.	Animal model	Wu A et al. 2006 [141]
Curcumin	Dose: 1.25, 2.5, 5, 10 mg/kg, intraperitoneally daily single dose	Depression: exerts antidepressant-like effects through the central monoaminergic neurotransmitter systems.	Animal model	Xu Y et al. 2005 [142]
Curcumin	Dose: 200 mg/kg, intraperitoneally daily for 7 days.	Brain ischemia: curcumin attenuated forebrain ischemia-induced neuronal injury and oxidative stress in hippocampal tissue.	Animal model	Al-Omar FA et al. 2006 [143]
Curcumin	Dose: 100, 200, 300 mg/kg, Orally, single dose	Epilepsy: Curcumin (300 mg/kg) significantly increased the latency to myoclonic jerks, clonic seizures as well as generalized tonic–clonic seizures and reduced oxidative stress and cognitive impairment	Animal model	Mehla J et al. 2010 [144]
Curcumin	Dose: 50 mg/kg, Orally, daily for 4 days	Parkinson’s Disease: curcumin protects the tyrosine hydroxylase-positive cells in the substantia nigra and dopamine levels in the striatum through its antioxidant capabilities	Animal model	Zbarsky V et al. 2005 [145]
Curcumin	Concentration 0–25 μM for 24 h	Parkinson’s Disease: these protective effects are attributed to the antioxidative properties also modulation of nuclear factor kappaB translocation.	In vitro	Wang J et al. 2009 [146]
Curcumin/its metabolite	Dose: 80 mg/kg, intraperitoneally, daily for 7 days	Parkinson’s Disease: curcumin and tetrahydrocurcumin reversed the MPTP induced depletion of dopamine and DOPAC through inhibition of MAO-B activity.	Animal model	Rajeswari A et al. 2008 [147]
Curcumin	Concentration 4 μM for 48 h	Parkinson’s Disease: curcumin could alleviate α-synuclein-induced toxicity, decreased ROS levels and protected cells against apoptosis.	In vitro	Wang MS et al. 2010 [148]
Curcumin	Concentration 0–1 μM for 2 times changing in 6 days treatment	Parkinson’s Disease: curcumin protects cells against A53T mutant α-synuclein-induced cell death through prevention of oxidative stress and the mitochondrial rescue	In vitro	Liu Zet al. 2011 [123]
Manganese complexes of curcumin	In vitro: 0–5 μg/mL for 3 h In vivo: 3 times (50 mg/kg × 3) at time points 1, 3, and 7 h post first MPTP sc injection, intraperitoneally	Neuroprotection: treatment with this complex attenuated MPTP-induced striatal dopamine depletion significantly	Animal model/In vitro	Vajragupta O et al. 2003 [149]

### 2.7. Protective Effect of Curcumin in Eye Diseases

Table 7 summarized some studies suggesting the efficiency of curcumin in eye diseases. All the studies were carried out on experimental models using curcumin except one in which researcher used nanoparticles. Curcumin beneficial effects have been proved for major eye diseases [150] such as glaucoma [151,152], age-related macular degeneration [153], diabetic retinopathy [154], cataract [155,156], corneal neovascularization [157], dry eye disease [158] and conjunctivitis [159]. In general, a wide variety of mechanisms have been raised for these effects of which antioxidative stress, anti-angiogenesis, anti-inflammatory and anti-apoptotic are more important [150].

## 3. Cellular and Molecular Targets of Curcumin

The cellular and molecular targets of curcumin have been summarized in Figure 2 [162,163,164,165]. It has been categorized based on the role of targets and the effect of curcumin on those. Curcumin clearly diminishes mRNA production of pro-inflammatory mediators, including cytokines and relative enzymes such as cyclooxygenase (COX)-2, and inducible nitric oxide synthase (iNOS) [166,167,168]. Apparently, this is due to it inhibitor effect of transcription factors such as activator protein (AP)-1 and nuclear factor (NF)-κB-mediated gene [143,169]; however, the direct molecular targets at low doses are not entirely clear.

There are some controversies regarding the effect of curcumin on these factors. To our knowledge, some of these effects are probably dose-dependent. For example, curcumin increase and decrease the apoptosis proteins and markers depending on its dose or concentration. In fact, in order to clarify the exact mechanism, further investigations using a wide range of doses are needed to explore the dose-dependent effects of curcumin. This hypothesis has been supported by the results of previous studies for instance, curcumin at low doses (<100 nM) prevents AP-1 and NF-κB-mediated transcription, which might rely on the inhibition effect on histone acetylase (HAT) or activation of histone deacetylase (HDAC) activity [170] while at high doses (>3 μM) curcumin can act as an alkylating agent in a study on colon cancer [171]. In fact, some of these curcumin effects at high doses in vitro are obviously toxic and beyond its usage in cancer therapy.

## 4. Curcumin Metabolism and Degradation

Following oral administration of curcumin, it is metabolized extensively. It has been reported that the phase I reduction reaction and phase II conjugation reaction were the major metabolic pathways of curcumin in animals, and dihydrocurcumin, tetrahydrocurcumin, octahydrocurcumin, and hexahydrocurcumin were the main metabolites in phase I [172].

There is some evidence indicating that metabolites are the major contributors of the pharmacological activity of curcumin. For instance, dihydrocurcumin diminished lipid accumulation, oxidative stress, and insulin resistance in oleic acid-induced L02 and HepG2 cells [173].

Tetrahydrocurcumin, another metabolite of curcumin, has been reported to show a wide range of therapeutic properties [174,175]. The properties of this metabolite are comparable to those of curcumin. However, studies revealed that Tetrahydrocurcumin is more potent than curcumin as an anti-inflammatory, antioxidant, neuroprotective agent and anti-cancer [174]. Several lines of evidence demonstrated that this curcumin metabolite has higher antioxidant activity and upregulates the antioxidant enzymes in different pathological conditions including atherosclerosis [176], diabetes [177], hyperlipidemia [178], and neurotoxicity [179]. Having additional hydrogen molecules, Tetrahydrocurcumin is more hydrophilic than curcumin [180] and pharmacokinetic assessments reveal that it is more stable than curcumin in 0.1 M phosphate buffers at neutral pH and plasma [181]. Moreover, Its half-life in in vitro studies is significantly longer than that of curcumin [182]. Together, these properties suggest that THC may be more potent and efficacious against human diseases than curcumin, owing to its distinct chemical properties and stability. Recent studies showed that curcumin quickly degrades in aqueous buffer and some degradation compounds are produced [183] comprising alkaline hydrolysis products (ferulic acid, vanillin, ferulaldehyde, and feruloyl methane) which are formed through ahydroxyl ion (OH-)-mediated hydrolysis reaction [184] and autoxidation products (bicyclopentadione), are formed through a radical-mediated process [185]. Evidence suggests that curcumin mostly degrades via oxidation pathway rather than through hydrolysis [183].

Although some of these degradation products are biologically active, they are noticeably less-active in comparison to curcumin [183], supporting the idea that chemical degradation of curcumin has a limited contribution to its biological and pharmacological activities.

## 5. Curcumin Toxicity

In addition to the wide variety therapeutic effects of curcumin discussed above, there are some reports about its toxic potential that could be discussed from 2 aspects; first, even in the therapeutic doses its reactivity against a number of enzymes which act through different cellular mechanisms, provides adverse effects. For instance, its glutathione S-transferase (GST) inhibition can lead to impaired detoxification and potential toxic drug−drug contraindications [186] or the human ether-a-go-go-related gene (hERG) channel inhibition which may lead to cardiotoxicity [187,188]. Second, high doses of curcumin have been reported to be toxic for cells inducing apoptosis [189]. In some studies of therapeutic utility, it has been shown as cytotoxic against a number of important cancer cell lines as mentioned before or even cytotoxic against normal human lymphocytes [190] and noncancerous cell lines [191].

Despite these effects, curcumin is known to be safe by Food and Drug Administration (FDA). Several studies including preclinical and clinical evaluated the safety of this compound [192,193,194]. The maximum proposed dose of curcumin varies, ranging from daily consumption of 3 mg/kg to 10 g [195]. In a clinical study, no serious adverse effect was observed in any of the healthy subjects who used a daily dose of 12 g [196].

According to the literature, curcumin didn’t show mutagenic and genotoxic effects and is safe in pregnant animals. However, more studies in humans are needed. In addition, animal studies didn’t report reproductive toxicity of curcumin following oral administration. Curcumin was safe even at the high doses: in a phase I human trial 25 subjects used up to 8000 mg of curcumin per day for 3 months without showing toxicity. Five other human trials using 1125–2500 mg of curcumin per day have also found it to be safe [197] as well as in another study in which it was used 6 g/day orally for 4–7 weeks.

In addition, a good safety profile has been reported for curcumin in patients with cardiovascular risk factors or pre-malignant lesions of internal organs who took a dose of curcumin ranging from 500 to 8000 mg/day for a 3 months period [192,198]. This safety has been observed in patients with different cancers, including colorectal (taking, ranging from 36 to 180 mg/day for up to 4 months), breast (taking up to 6000 mg/day) and pancreatic cancer (taking 8000 mg/day of curcumin for 2 months) [86,199,200].

Having in mind that the majority of studies reporting curcumin safety has been performed for short periods of time so far and, there is not complete evidence regarding the consequences of chronic administration of this compound. Therefore, trials and more studies are needed specially on novel formulations and in long term use to find out all aspects of toxicity of curcumin.

However, only limited adverse effects such as gastrointestinal upsets have been stated in human. In addition, abdominal pain has also been reported followed by curcumin consumption at a dose of 8000 mg/day in patients with pancreatic cancer [201]. These gastrointestinal side effects might be related to the curcumin-induced COX inhibition and the subsequent inhibition of prostaglandin (PG) synthesis and/or its influence on the composition of the gut microbiome, which needs further study to be established. In addition to gastrointestinal side effects, a mild headache or nausea have been reported in few patients affected by primary sclerosing cholangitis taking curcumin in dose of up to 1400 mg/day [202].

Moreover, recent reports of liver diseases related to curcumin attracted the medical community’s attention to its possible hepatotoxicity [203]. Whether this effect belongs to curcumin molecule or other possible contamination has to be elucidated.

Evaluating the other routes of administration, in a short-term intravenous dosing of liposomal formulation, curcumin has been shown to be safe up to a dose of 120 mg/m^2^ on healthy subjects in a clinical trial, while in a dose escalation study in patients with metastatic cancer a dose of 300 mg/m^2^ over 6 h reported to be the maximum tolerated dosage [204,205]. Besides, one case of hemolysis and one death associated with intravenous curcumin preparations were reported, indicating that regarding the safety of intravenous administration of curcumin, further studies and data are required [205,206,207].

## 6. Pharmacokinetic Deficiency of Curcumin and Current Attempts

As described previously, the major constituent of extracts of *C. longa* are called curcuminoids, which includes curcumin demethoxycurcumin, and bisdemethoxycurcumin, along with numerous and less abundant secondary metabolites [208]. According to our literature review, many in vitro studies have used synthetic curcumin, while most animal studies and clinical trials used the curcuminoids mixture. In fact, these compounds are completely different in pharmacokinetic parameters.

Gastrointestinal absorption and bioavailability are two most important and critical characteristics in the pharmacokinetics of any compound. From the preclinical and clinical studies, we have found that curcumin is poorly absorbed following oral administration. It has been reported that oral bioavailability of curcumin was only 1% and the highest amount of plasma concentration was 0.051 μg/mL from 12 g curcumin in human, 1.35 μg/mL from 2 g/kg in rat, and 0.22 μg/mL from 1 g/kg in mouse [196]. Pharmacokinetic assessments showed that there were just negligible quantities detected in liver and kidney (<20 μg/tissue) following curcumin oral administration. These studies indicated the nano-molar plasmatic concentration of this compound, with limited biological effects [209,210,211,212,213].

Several studies, including clinical trials, have been performed using curcumin in a wide variety of oral formulations. From low oral doses to high amount of 12 g/day have been given and tested [196]. Data shown in Table 8 summarize some of formulations and their pharmacokinetic properties. In order to increase curcumin bioavailability, different pharmaceutical strategies, including combination with adjuvant substances such as piperine, encapsulation, formulation in nanoparticles, liposome, micelles, nanomicellizing solid dispersion etc., have been proposed, shedding new light to overcome this pivotal limitation [214,215,216,217].

As presented below, there is a remarkable difference between the amount of plasma level of curcumin in different products. Interestingly, new formulations including nanoparticles, liposomes, and micelles exert a higher amount of C-max, the maximum (or peak) serum concentration that a drug achieves, compared to the other formulations (Table 8).

Based on the Nutraceutical Bioavailability Classification Scheme (NuBACS), curcumin exerts poor bio-accessibility, due to its low solubility in water and low stability [218].

It is well documented that water-solubility has a crucial role in oral absorption of compounds. Pharmacokinetics assessments revealed that curcumin is a poorly water-soluble agent (about 11 ng/mL) [219] and considerable efforts have been made to improve this characteristic.

Another important point that has to be taken into account in pharmacokinetic of curcumin, is its susceptibility to degradation, which has been shown to be pH-dependent. Indeed, under the alkaline conditions (pH > 7), curcumin degrades to Trans-6-(40-hydroxy-30-methoxyphenyl)-2, 4-dioxo-5-hexanal, ferulic acid, feruloylmethane, and vanillin within 30 min. while, in acidic medium, curcumin degradation is much slower, with less than 20% of total curcumin at 60 min [220,221].

Following oral administration, the major portion of curcumin is excreted through the feces while a small portion, which is absorbed within the intestine, rapidly metabolized in the livers and plasma [222]. Curcumin is extensively converted to its water-soluble metabolites (glucuronides and sulfates) and excreted through urine [222,223].

Curcumin also undergoes extensive hepatic first-pass metabolism through glucuronidation and sulfation, with the metabolites that exerts a remarkably lower biological function in comparison with parent curcumin [214]. Taking advantage of extensive research on curcumin metabolism, a curcumin-converting enzyme named “Nicotinamide adenine dinucleotide phosphate (NADPH)-dependent curcumin/dihydrocurcumin reductase” has been known which has purified from *E. coli*, clarifying the role of human intestinal microorganisms in the metabolism of curcumin in vivo [224].

The table provided below shows the studies on curcumin formulations, routes of administration, dose Plasma/tissue level (Cmax) and the time to maximum concentration (Tmax, min) in animals and humans. There is a significant variation in the doses used from the formulations included in this review as well as their Cmax and Tmax were really different.

Having two phenolic hydroxyls and one enolic hydroxyl group, which can form hydrogen bonds with phospholipids polar groups, in one formulation named Meriva, curcumin has complexed with phosphatidylcholine, and this structure protects it from degradation and augmenting the cellular uptake across lipophilic cell membranes by facilitated diffusion mechanism [225,226]. Taking advantage of this formulation, curcumin absorption and bioavailability were significantly increased compared to unformulated curcumin in a randomized, double-blind, study in human [225]. The Tmax has been reported 15 to 30 min, which showed a considerable increase in speed of absorption compared to the other formulations.

In another compound named LongVida^®^, curcumin is included in a pharmaceutical form of solid lipid particle (SLP)-based formulation. This complex keeps the curcumin from rapid degradation and excretion so improving the systemic curcumin plasma concentration and half-life and eventually improved bioavailability compared to unformulated curcumin. In a clinical trial, oral administration of 650 mg in human exerts 22.43 ng/mL plasma level, which was shown to be higher than curcuma extract and unformulated curcumin [227].

There are also some micronized formulations for curcumin, which were reported to have 9-fold increased bioavailable than unformulated. The micronized curcumin has a smaller diameter of drug particles, which increases the surface area to drug ratio and finally augmentation in the dissolution rate that directly increase the bioavailability of curcumin [228]. Interestingly in a human study, administration of 500 mg of a micronized formulation exerts a considerable 0.60 μg/mL plasma level, which was a promising results for this compound [229].

One of the early and basic strategies for increasing the bioavailability of curcumin was its combination with other compounds that could change its characteristics; among them piperine was known to improve the oral absorption of curcumin in humans. It is a P-glycoprotein inhibitor and enhances the curcumin absorption in the intestine by reducing the efflux phenomenon [230].

P-glycoprotein (P-gp) is a drug transporter that effluxes drugs and other foreign substances out of cells from gastrointestinal tract, brain, liver, and kidney and exerts an important role in drug pharmacokinetics and pharmacoresistance. Piperine has been reported to enhance the bioavailability of curcumin, as a substrate of P-gp by at least 2000% [231].

Piperine also inhibits uridine diphosphate-glucuronosyltransferase (UGT) thus increasing the curcumin availability in the systemic circulation. As shown in Table 8, adding piperine 20 mg/kg to curcumin 2 g/kg in rats exerts a C-max of 1.55 µg/mL in animals following oral administration [231].

In another attempt, micellization technique was employed to improve the solubility of curcumin in a compound named NovaSol^®^. In this formulation, curcumin incorporated in a nonionic surfactant Tween 80, caused the formation of liquid micelles, which increased dissolution and improved its absorption. This compound has been studied in a single-blind crossover study in healthy subjects and interestingly they showed a 185 folds higher than that of unformulated curcumin [232].

As previously described in this review, nanoparticle-based formulations have been extensively tried to improve the bioavailability and reduction of adverse drug reactions. In this regard, a colloidal nanoparticle dispersion of curcumin has been produced and its pharmacokinetic characteristic has been studied in humans. This formulation which is called Theracurmin™ has been showed to have relative bioavailability almost 16 time as compared to unformulated curcumin. In fact, the colloidal nanoparticle dispersion improves the solubility of curcumin and its oral bioavailability in healthy volunteers. In one study on this formulation, authors stated that two factors including improved solubility by adding a compound named “gum ghatti” and reduced particle size improved the clinical bioavailability of Theracurmin™ [233].

**Table 8 ijms-21-06619-t008:** Pharmacokinetic characteristics of some curcumin formulations in animal and clinical studies.

Product	Species	Route of Administration	Dose	Plasma/Tissue Level (Cmax)	Time to Maximum Concentration (Tmax) min	Ref.
Curcuminoids	Rat	Oral	500 mg/kg	0.06 µg/mL	41.7	[234]
Curcumin	Rat	Oral	200 mg/kg	1.2 µg/mL	no	[235]
Curcumin & Curcumin phospholipid complex (Meriva)	Rat	Oral	340 mg/kg	6.5 nM & 33.4 nM	30 & 15	[236]
Curcuminoids	Human	Oral	450–3600 mg	10 nM/g tissue	No data	[237]
Curcumin	Human	Oral	3600 mg	12.7 nmol/g tissue	No data	[238]
Curcuminoids	Rat	Oral	100 mg/kg	trace	60	[239]
Curcumin	Rat	Oral	400 mg	trace	No data	[209]
Curcumin	Mouse	Intraperitoneal	100 mg/kg	trace	No data	[240]
Curcumin	Human	Oral	3600 mg	10 nM	No data	[85]
Curcumin	Human	Oral	1200 mg	51 ng/mL	No data	[196]
Curcumin	In vitro	Exposure	5–75 µg/mL	3% in tissue	No data	[241]
Curcumin	Rat	Oral	10, 80, 400 mg	65–66%	No data	[242]
Curcuma extract	Human	Oral	440 and 2200 mg/day	175 to 310 μg/L	No data	[199]
Phospholipid formulation	Human	Oral	200−300 mg	50 ng/mL	240	[225]
Solid lipid curcumin particle	Human	Oral	650 mg	22.43 ng/mL	160	[227]
Curcumin-impregnated soluble dietary fiber dispersions	Human	Oral	600 mg	0.37 μg/g tissue	60	[243]
Micronized formulation	Human	Oral	500 mg	0.60 μg/mL	No data	[229]
Micronized formulation	Human	Oral	500 mg	50.6 nM	460	[232]
Liquid micelles formulation	Human	Oral	500 mg	3701 nM	66	[232]
Curcumin/piperine co-administration	Rat	Oral	Curcumin 2 g/kg piperine 20 mg/kg	1.55 µg/mL	120	[231]
Lipophilic matrix	Human	Oral	376 mg	18 ng/mL	60	[244]
γ-cyclodextrin complex	Human	Oral	376 mg	87 ng/mL	60	[244]
Colloidal nanoparticle	Human	Oral	30 mg	29.5 ng/mL	60	[233]
Curcumin	Rat	Intravenous	40 mg/kg	No data	No data	[245]
BCM-95^®^ CG A patent formulation	Human	Oral	2000 mg	456.88 ng/g tissue	206	[246]

## 7. Conclusions and Future Prospects

Similar to any drug, curcumin still requires extensive preclinical and clinical assessments to clarify its pharmacokinetics, dosing, and potential toxicity, the issues that are of essential concern of drug companies.

According to the therapeutic effects that were summarized in this review, curcumin plays different roles through apposite mechanisms such as cytotoxic and cytoprotective role in different disease. For understanding this controversy and diversity of effects, it is crucial to pay attention to the doses and concentrations that have been used in different situations. It seems that curcumin’s behavior changes in various doses and conditions, and this issue should be highly considered in clinical trials, especially in new formulations with higher bioavailability characteristics. Confirming this idea about the dose-dependent effect of curcumin, recently, we showed that low concentration of curcumin (5 µM) exerts cytoprotective effect in SH-SY5Y neuroblastoma cells while higher concentration (15 µM) induced apoptosis and cell death [126].

Despite having a wide range of pharmacological activity, there is still a key question on curcumin to be answered, that is, why it remains as a supplement and couldn’t play a higher-order role as a drug in treatment of diseases?

In response to this essential concern, we believe that pharmacokinetic deficiency is the main obstacle and studies to address this issue should be more considered before doing further preclinical and clinical studies. Confirming this idea, researchers have discussed that given its unstable, reactive, and poor bioavailability nature, further clinical trials on curcumin are unwarranted [34,191]. Therefore, this decade has witness an escalation in curcumin formulations addressing its pharmacokinetics (Figure 3).

A wide variety of compounds have been formulated and tried to improve the pharmacokinetic problems, especially absorption, which is the first step in oral administration of drugs.

Although the absorption and bioavailability are the crucial issues, curcumin distribution in body tissues should be take into account for its biological activity. In fact, there are just limited number of investigations have addressed this problem so far. In this regard, researchers reported that oral administration of curcumin (400 mg) in rats exerts only traces of unchanged drug in other organs such as liver and kidney [209].

Novel delivery strategies including nanoparticles, liposome, phospholipid complex, micro-emulsions, or polymer micelles suggest significant promise and are worthy of further investigation to enhance the oral bioavailability of curcumin. Here we reviewed some new formulations on the market, and some of their pharmacokinetic characteristics have been compared in Table 8. Most clinical studies on these new formulations were conducted randomized and blinded (neither researchers nor volunteers were aware of the treatment). However, the most important limitation of these investigations was the low number of subjects were recruited in the studies. In addition, there are remarkable differences in the sampling time following oral administration of curcumin that may affect the pharmacokinetic parameters including Tmax and Cmax. In fact, a wide-ranging of variability in this research, including route of administration, study design, sample size, and even the type of reports, makes it very difficult to directly compare and conclude which formulation is the best in pharmacokinetic characteristics overall. However, an overall view to these results shows that novel formulations involving nanoparticles and micronized products showed improved solubility and a relatively more amount of blood concentration following oral administration compared to the others.

With regards to the fact that most of the investigations reported in this review have been designed and performed in vitro, extensive harmonized large clinical study including different formulations in humans with the same design will make great sense and is needed to compare the bioavailability and efficacy of these curcumin formulations.

Considering the pharmacokinetic properties of curcumin metabolites and their beneficial effects, such as regulation of oxidative stress, prevention of neurodegenerative disorders and their potential mechanism to prevent or treat human diseases, developing new formulations and performing clinical trials using these compounds is also suggested.

Finally, bypassing the limitations of curcumin mentioned in this review, such as low oral bioavailability, distribution, and metabolism, will pave the way to do more applicative research and clinical studies on curcumin as a drug in the near future.

## Figures and Tables

**Figure 1 ijms-21-06619-f001:**
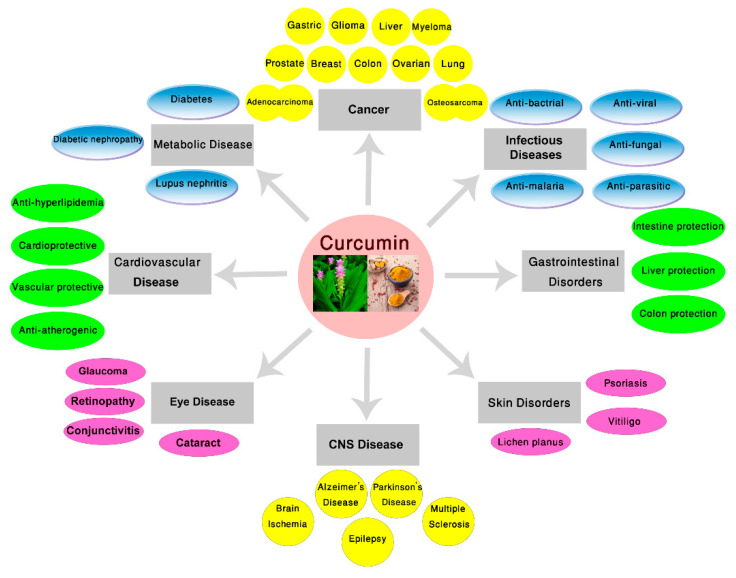
Effect of curcumin in different diseases. The therapeutic benefits obtained from in vitro cell cultures to small and large animal studies as well as clinical trials. CNS: Central Nervous System.

**Figure 2 ijms-21-06619-f002:**
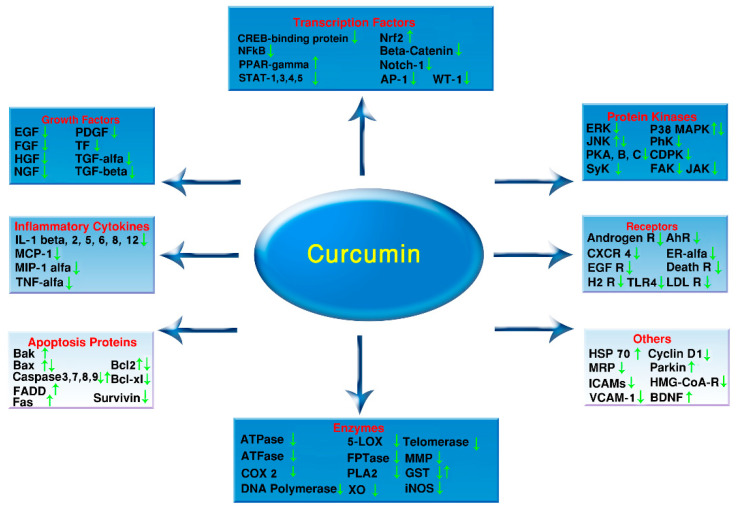
Cellular and molecular targets of curcumin. Curcumin directly or indirectly interacts with numerous molecular targets and modulates their activity and function. **AH R:** Aryl hydrocarbon receptor, **AP-1:** Activator protein 1, **Bax:** Bcl-2-associated X protein, **BDNF:** Brain-derived neurotrophic factor, **CDPK:** Calcium-dependent protein kinases **CRDB:** Curcumin Resource Database, **CREB:** cAMP response element-binding protein, **COX-2:** Cyclooxygenase-2, **CXCR 4:** C-X-C Motif Chemokine Receptor 4, **EGF:** Epidermal growth factor, **ER-alfa:** Estrogen receptor alfa, **ERK:** Extracellular signal-regulated kinases, **FADD:** Fas Associated via death domain, **FAK:** Focal adhesion kinase, **FAS:** Fas cell surface death receptor, **FGF:** Fibroblast growth factors, **GST:** Glutathione-S-transferase, **HAT:** Histone acetylase, **H2 R:** Histamine H2 receptor, **HDAC:** Histone deacetylase, **HGF:** Hepatocyte growth factor, **HMG-CoA-R:** 3-hydroxy-3-methyl-glutaryl-CoA reductase, **HSP-70:** Heat shock protein 70, **IBD:** Intestinal inflammatory diseases **ICAMs:** Intercellular cell adhesion molecules, **IL:** Interleukin, **iNOS:** Inducible nitric oxide synthase, **JAK:** Janus kinase, **JNK:** c-Jun N-terminal kinases, **LDL R:** Low-Density Lipoprotein Receptor, **MCP-1:** Monocyte chemoattractant protein-1, **MIP-1α:** Macrophage inflammatory proteins, **MMP:** Matrix metallopeptidases, **MRP:** Multidrug resistance-associated protein, **NFκB:** Nuclear Factor kappa-light-chain-enhancer of activated B cells, **NGF:** Nerve growth factor, **Nrf2:** Nuclear factor erythroid 2–related factor 2, **P38-MAPK:** P38 mitogen-activated protein kinases **PDGF:** Platelet-derived growth factor, **PhK:** Phosphorylase kinase, **PKA:** Protein kinase A, **PLA2:** Phospholipase A2, **PPAR-gamma:** Peroxisome proliferator-activated receptor gamma, **ROS:** Reactive oxygen species, **RNS:** Reactive nitrogen species, **SLP:** Solid lipid particle, **STAT:** Signal transducer and activator of transcription, **SyK:** Spleen tyrosine kinase, **TF:** Tissue factor, **TGF-α:** Transforming growth factor alpha, **TGF-β:** Transforming growth factor beta, **TLR:** Toll-like receptors, **TNF-α:** Tumor necrosis factor alpha, **UGT:** Uridine diphosphate-glucuronosyltransferase, **VCAM:** Vascular cell adhesion molecule, **XO:** Xanthine oxidase, **5-LOX:** 5-Lipoxygenase.

**Figure 3 ijms-21-06619-f003:**
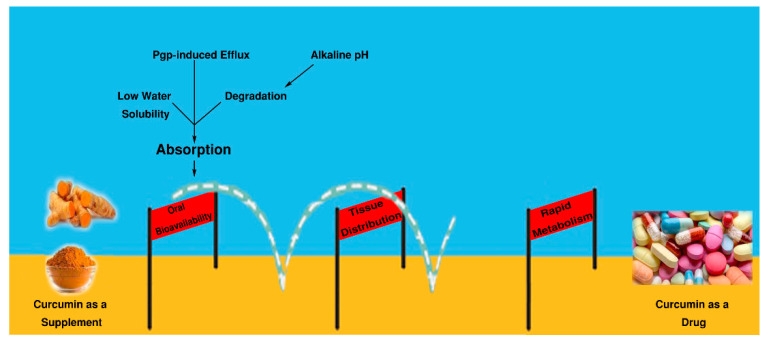
Obstacles against the marketing of curcumin as a drug. Challenges in curcumin oral bioavailability, distribution and metabolism, the main pharmacokinetic parameters, that emerged as major obstacles limiting the therapeutic efficacy and marketing of curcumin as a drug.

**Table 7 ijms-21-06619-t007:** Effect of Curcumin in eye diseases.

Product	Dose or Concentration Used	Effect and Findings	Type of Study	Studied by
Curcumin	In vitro: 0.1, 1, 10 μM for 2 for 1 h In vivo: 10 mg/kg orally daily for 6 weeks	Glaucoma: curcumin, increased the cell viability and decreased intracellular ROS and apoptosis significantly. In vivo study, curcumin protected rat BV-2 microglia from death significantly	In vitro/Animal model	Yue YK et al. 2014 [151]
Curcumin	Curcumin (0.01%, 0.05% and 0.25%, which are equivalent to 100, 500 and 2500 ppm in diets) for 2 days before the injury	Glaucoma: Curcumin protected retinal neurons and microvessels against Ischemia/Reperfusion injury through inhibition of injury-induced activation of NF-κB and STAT3, and on over-expression of MCP-1.	Animal model	Wang L et al. 2011 [152]
Curcumin	Concentration 0–100 μM for 24 h	Age-related macular degeneration: Curcumin improved cell viability and reduced apoptosis and oxidative stress and had a significant influence on expression of apoptosis-associated proteins and oxidative stress biomarkers.	In vitro	Zhu W et al. 2015 [153]
Curcumin	Dose: 1 g/kg orally, daily for 16 weeks	Diabetic retinopathy: curcumin positively controlled the antioxidant system, pro-inflammatory cytokines, tumor necrosis factor-α and vascular endothelial growth factor in the diabetic retinae	Animal model	Gupta SK et al. 2011 [154]
Curcumin + sodium selenite	Concentrations Curcumin 100, 200 μM sodium selenite 100 μM	Cataract: Curcumin suppressed selenium-induced oxidative stress and cataract formation through preventing depletion of antioxidants, and inhibiting generation of free radicals, and by inhibiting iNOS expression	In vitro	Manikandan R et al. 2009 [155]
Curcumin and Turmeric extract	0.002%–0.01% curcumin and 0.5% turmeric in diet	Cataract: turmeric and curcumin were effective against the diabetic cataract development in rats.	Animal model	Suryanarayana P et al. 2005 [156]
Curcumin nanoparticles (NP)	In vitro: curcumin 5–20 μM for 24 h In vivo: 20-μL solution containing 80 μg curcumin for 14 days	Corneal neovascularization: NP increased the retention of curcumin in the cornea and suppressed the expression of VEGF, inflammatory cytokines, and MMP so prevented corneal neovascularization through suppressing the NFκB pathway.	In vitro/Animal model	Pradhan N et al. 2015 [157]
Curcumin	Concentrations 1–30 μM for 24 h	Dry eye disease: Curcumin has the potential for dry eye disease. It prevented the hyperosmoticity-induced increase of NF-κB and IL-1β production	In vitro	Chen M et al. 2010 [158]
Curcumin	Dose:10, 20 mg/kg intraperitoneally twice on days 14 and 17, beginning 1 h before the challenge in the conjunctival sac	Conjunctivitis: curcumin suppressed the allergic conjunctival inflammation in an experimental model.	Animal model	Chung SH et al. 2012 [159]
Curcumin	Dose: 2.5 and 10 μM) injected into the vitreous of C57BL/6 mice.	Retinal degeneration: curcumin attenuated retinal ganglion cell and amacrine cell death by restoring NF-κB expression.	Animal model	Burugula B et al. 2011 [160]
Curcumin Nanoparticle	In vitro: curcumin 0–20 μM for 24 h In vivo: topical eye drop daily for 21 days	Neuroprotective in eye disease: Curcumin-loaded nanocarriers protected a retinal cell line against glutamate and hypoxia-induced injury	In vitro/Animal model	Davis BM et al. 2018 [161]

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
