# Peer review of "Obstacles against the Marketing of Curcumin as a Drug"

_ijms, 2020, doi:10.3390/ijms21186619_

Round 1

Reviewer 1 Report

Please, revise the tables adding chemical characteristic of the compounds and dosages.

Please, revise the discussion on new formulations. Some recent papers on curcumin metabolism are missing. Also efficacy clinical trials on new formulations should be discussed.

Please, revise the English language.

Author Response

Thank you for your valuable comments.

  1. Please, revise the tables adding chemical characteristic of the compounds and dosages.

Response: All tables have been revised, some other information including the chemical characteristics, dosage and main findings of studies have been added.

  1. Please, revise the discussion on new formulations. Some recent papers on curcumin metabolism are missing. Also efficacy clinical trials on new formulations should be discussed.

Response:

The section of new formulations has been revised and some points regarding their characteristics and efficacy of new formulations have been added to the discussion and conclusion part. “Novel delivery strategies including …………..are suggested to be tried.”

A section entitled “Curcumin metabolism and degradation” has been added to the manuscript and some characteristics of curcumin metabolites have been notified. In addition, a paragraph has been added to the discussion “Considering the pharmacokinetic properties of curcumin metabolites which described before and their beneficial effects, such as regulation of oxidative stress, prevention of neurodegenerative disorders and their potential mechanism to prevent or treat human diseases, developing new formulations and performing clinical trials using these compounds is also suggested.”

  1. Please, revise the English language.

Response:

The manuscript has been totally revised; spelling and grammatical errors have been corrected.

Reviewer 2 Report

The review "Obstacles Against the marketing of Curcumin as a Drug
by Kambiz Hassanzadeh1,2,#, Lucia Buccarello1
, Jessica Dragotto1, Asadollah Mohammadi2 4 , Massimo
Corbo3 and Marco Feligioni* 5 1,3" is a well written and timely article. Authors highlighted the scientific evidence on the basic and clinical effects and molecular targets of curcumin. Moreover, authors also discuss its pharmacokinetic and problems for marketing curcumin as a drug. I think current article can be accepted in it's current form.

Author Response

Thank you for accepting our manuscript in the current format for publication.

Reviewer 3 Report

The manuscript describes the current information about curcumin's therapeutic effect. The very low bioavailablity is the limitation for being as a drug. However, there is no novel concept of how to overcome the limitations of curcumin. I do not think the manscript will challenge the readers.

Author Response

Dear Reviewer

Thank you for your valuable comments.

  1. There is no novel concept of how to overcome the limitations of curcumin.

Response: As we described in the introduction and discussion part of this study, there are a wide variety of papers in the literature, more than 10,000 papers, in PubMed with the title of curcumin. Despite having a wide range of pharmacological activity, there is still a key question on curcumin to be answered, that is, why it remains as a supplement and couldn’t play a higher-order role as a drug in treatment of diseases?

This key question should be answered and researchers should focus on solving the obstacles against curcumin. In fact, this review indicates the pharmacological effects, molecular targets and finally discussing the problems that should be addressed by researchers so that the usage and application of curcumin as a drug in market will increase. From this point of view there is not a paper in literature so we believe that gathering all these data in the format of a review article could be useful for increasing and updating our knowledge on curcumin problems and try to overcome its limitations.

Reviewer 4 Report

The authors should get editing help from someone with full professional proficiency in English.

Title: Marketing should also be in uppercase

Abstract

Line 16: Derived from what?

Line 20: Curcuma longa should be in italic

Line 22: Replace it for its

Line 27: Replace rout for route

Line 27: Replace “attempt” for “attempted”

Introduction and History

Line 42/47/62: Curcuma longa should be in italic

Line 42-44: revise the sentence “people of the Indian”

Line 49: Please explain what you mean with curcumin is the key colored compound. Do you mean key therapeutic compound?

Line 49: “It is the”, which one? Curcumin? It is not clear.

Line 58: Remove “that are”

Line 59: Add reference number from Kumar and colleagues (2015)                                                                                                                                       

Line 65: Replace are for is

Line 75: “including cells” - what the authors mean with this?

Figure 1: The figure is a bit blurred. The figure quality should be improved. The word “organ” should be removed from caption. Add the meaning of CNS to the figure caption.

Section “Disease targets of curcumin: from cell lines to clinical trial studies” – This paragraph (line 82-86) does not discuss anything. Bibliography is not cited and information is not given. Figure 1 is referenced but it is not enough to replace the information that should be in this paragraph. After this paragraph the authors move to a different topic that according to the letter size is not inside the previous topic. It should at least be an indication that the following topics are related to this section (or they are not?).

Line 96-98: Review sentence

Line 99: Remove first “of”. Replace “viruse’s” for viral.

Line 101: Add PI3K/Akt, NF-κB meaning.

Line 102: Remove “has”

Line 106: Escherichia coli should be in italic.

Line 110: “Herein” – Where? Where is the overview? Are you referring to Table 1?

Table 1: Table 1 is very incomplete. It must contain a column with the relevant findings in each study otherwise is insignificant. It should be explained why the study is relevant. Where it says “Essential oil” should be added the amount of curcumin present in the essential oil. What the authors mean with “Methanol extract”? What is the purity of curcumin used in these studies? The extracts still contain bisdemethoxycurcumin and demethoxycurcumin right? Add CT meaning (clinical trial).

Line 117: Curcumin should be in lower case

Line 127: Replace “hapatoprotective” for “hepatoprotective”

Line 131: Remove dot after table 2

Line 131: What do you mean with the data is adapted?

Table 2 (and 3-7): Table 2 (and 3-7) is very incomplete. It must contain a column with the relevant findings in each study otherwise, for example curcumin mechanism of action. It should be explained why the study is relevant. In the tables 3, 4 and 6 there are reviews. However, original research studies should be cited instead.

Line 140: The paragraph here does not seem correct since the authors are still discussing ulcerative colitis.

Line 145: Add that before contribute

Line 147: Replace the fist has for have

Line 148: Different from what? Do you mean that the effects vary?

Line 155: Remove dot after table 2

Lines 182-186: The type of letter is bigger than in the remaining document.

Line 189: Nelumbo nucifera should be in italic

Line 190: Curcuma longa should be in italic, remove the L since it was not added before. In addition, after mentioning the first time the authors should write “C. longa” in italic.

Line 190: Replace provide for provided

Line 195: Add of after augmentation

Table 3: Explain what Tetrahydro-curcumin is.

Some of the studies in the tables are very old. It would be important to find more recent studies (last 5 years) for each disease.

Line 207/210: use anti-cancer or anticancer, not both.

Line 222: Replace reveals for reveal.

Line 225: Remove a before low levels.

Line 234: Define EGFR.

Line 234: An or In a clinical trial?

Line 236: Patients a?

Line 246 (364, 419): Replace rout for route.

Line 261: Add the word disease

Line 281: Define SUMO and JNK

Table 6: explain what Curcumin C3 complex is and specify curcumin concentration and other curcuminoids.

Tables in general: It is only referred CT. Are the authors referring only clinical trials phase 1? Or is there phase 2? It should be specified.

Line 296: Replace this for these

Figure 2: The figure is a bit blurred. The figure quality should be improved. All abbreviations should be defined in the figure caption (VCAM, TNF, EGF….)                                                                                                                                       

Lines 301-302: Please revise this sentence.

Line 350: Remove and

Line 362: Remove the

Line 365/366: in m2 the 2 should be elevated (correct every time is used).

Line 372: Major should be in lower case. Replace curcuma longa for C. longa (in italic).

Line 372-375: This is a repetition. It is already present in the beginning of the paper (lines 47-49).

Line 384: Remove a

Line 390: Remove are

Line 416: Define NADPH

Line 417: Replace Escherichia coli for E. coli (in italic)

Line 437-438: Revise sentence.

Line 439: Add which before increases

Table 8: This table needs a formatting revision (e.g. lower and upper cases). What do you mean with no in time maximum concentration? Why ref 249 has two different times? What is the unit of 33.4? Why using nM and in the other cases nmol/L? In ref 97 and 252, it is 3.6 or 3600 or 3 and 600? In ref 256 the authors presented 10,80,400 (are these 3 doses?). In ref 209 they wrote 1200, not 1,200…In ref 212 replace liter per L. This is confusing. Ref 257 and 261, g of what? Tissue? Ref 245, it should be mL instead of ml (correct other cases in the manuscript). Last reference from page 21 is not ok

Line 475: Replace has for were

Lines 500-504: Please revise the paragraph and pay special attention to expressions such as “need to be studied more”, “limit number of investigates” and “in other organs” (what were the other organs?).

Line 509: Revise sentence

Line 511: Population of objects??

Line 525: Add author contributions.

References: The manuscript has 261 references and several are quite dated. Primary research from the past 5 years should be prioritized.

Author Response

Reviewer 4

Comments and Suggestions for Authors

Title: Marketing should also be in uppercase. Response: It has been corrected “Obstacles Against the Marketing of Curcumin as a Drug”

Abstract

Line 16: Derived from what? Response:It has been corrected “phytochemicals to prevent”

Line 20: Curcuma longa should be in italic. Response:It has been corrected “Curcuma longa

Line 22: Replace it for its. Response:It has been corrected “its interesting pharmacological activities”

Line 27: Replace rout for route. Response:It has been corrected. “is given via the oral route”

Line 27: Replace “attempt” for “attempted”. Response: It has been corrected “researchers have attempted”

Introduction and History

Line 42/47/62: Curcuma longa should be in italic. Response:they have been corrected “Curcuma longa

Line 42-44: revise the sentence “people of the Indian”. Response:It has been revised “has been used by the Indian people for centuries with no known side effects”

Line 49: Please explain what you mean with curcumin is the key colored compound. Do you mean key therapeutic compound? Response:It means that among the 3 main curcuminoids, curcumin (75–80%), which is the main colored compound and the yellow color of this compound depends on the presence of curcumin.

Line 49: “It is the”, which one? Curcumin? It is not clear. Response: It has been corrected “Curcumin is the yellow pigment”

Line 58: Remove “that are”. Response: “that are” has been removed “Scientific documents are still published focusing on”

Line 59: Add reference number from Kumar and colleagues (2015) Response: the reference has been added. “In 2015, Kumar and colleagues [7] created”

Line 65: Replace are for is. Response:It has been corrected “Turmeric paste is also available in”

Line 75: “including cells” - what the authors mean with this? We mean Cellular studies. Response:It has been corrected. “in this paper we review several aspects including cellular studies to clinical trials related to curcumin”

Figure 1: The figure is a bit blurred. The figure quality should be improved. The word “organ” should be removed from caption. Add the meaning of CNS to the figure caption. High quality version of figure has been uploaded. word “organ” should be removed and the meaning of CNS has been added to the figure caption.

“Figure 1. Effect of curcumin in different disease. The therapeutic benefits obtained from in vitro cell cultures to small and large animal studies as well as clinical trials. CNS: Central Nervous System.”

Section “Disease targets of curcumin: from cell lines to clinical trial studies” – This paragraph (line 82-86) does not discuss anything. Bibliography is not cited and information is not given. Figure 1 is referenced but it is not enough to replace the information that should be in this paragraph. After this paragraph the authors move to a different topic that according to the letter size is not inside the previous topic. It should at least be an indication that the following topics are related to this section (or they are not?).

Response: This little paragraph is a short introduction for the other subtitles. We have better clarify this by changing the first sentence and also by changing the forma of the subtitles.” In this section we will discuss the several effect of curcumin including antimicrobial, gastrointestinal, cardiovascular, anti-cancer, ant-inflamatory, neuroprotective in different disorders which have been previously reports.”

Line 96-98: Review sentence. Response:It has been revised and corrected “Inhibition of assembly dynamics of FtsZ in the Z-ring may suppress the bacterial cell proliferation as one of the probable curcumin antibacterial mechanisms of action”

Line 99: Remove first “of”. Replace “viruse’s” for viral. Response:It has been corrected “plays an inhibitory role against numerous viral infection”

Line 101: Add PI3K/Akt, NF-κB meaning. Response: They have been added “such as phosphatidylinositol-3-kinase (PI3K)/Akt, nuclear factor kappa-light-chain-enhancer of activated B cells (NF-κB)”

Line 102: Remove “has”. Response: It has been removed “curcumin in last decade focused”

Line 106: Escherichia coli should be in italic. Response: It has been corrected “Escherichia coli

Line 110: “Herein” – Where? Where is the overview? Are you referring to Table 1? Response: It has been corrected “In table 1, there is an overview of the multipotent antimicrobial character”

Table 1: Table 1 is very incomplete. It must contain a column with the relevant findings in each study otherwise is insignificant. It should be explained why the study is relevant. Where it says “Essential oil” should be added the amount of curcumin present in the essential oil. What the authors mean with “Methanol extract”? What is the purity of curcumin used in these studies? The extracts still contain bisdemethoxycurcumin and demethoxycurcumin right? Add CT meaning (clinical trial).

Response:

All valuable comments and corrections raised by Reviewer regarding this table have been corrected. Some other information including the chemical characteristics, dosage and main findings of studies have been added. The amount of curcumin has been added in detail in a column. The comments on extracts have been revised and corrected. CT has been changed to clinical trial.

Line 117: Curcumin should be in lower case. Response: It has been corrected “activity of curcumin”

Line 127: Replace “hapatoprotective” for “hepatoprotective”. Response:It has been corrected “support the hepatoprotective effect of curcumin”

Line 131: Remove dot after table 2 Response:Table 2 Gastrointestinal Effect of Curcumin”

Line 131: What do you mean with the data is adapted? Response: It has been changed to “the data is mostly obtained from animal studies”

Table 2 (and 3-7): Table 2 (and 3-7) is very incomplete. It must contain a column with the relevant findings in each study otherwise, for example curcumin mechanism of action. It should be explained why the study is relevant. In the tables 3, 4 and 6 there are reviews. However, original research studies should be cited instead.

Response:

All valuable comments of Reviewer regarding tables (2-7) have been corrected. Some other information including the chemical characteristics, dosage and main findings of studies have been added. Review articles have been removed.

Line 140: The paragraph here does not seem correct since the authors are still discussing ulcerative colitis. Response: We have now changed the format of this paragraph which has now been included in the previous paragraph in order to join the ulcerative colites.

Line 145: Add that before contribute. Response:“that” has been added “possible mechanisms that contribute”

Line 147: Replace the fist has for have. Response:it has been corrected “microbiome have been widely studied”

Line 148: Different from what? Do you mean that the effects vary? Response:It has been rephrased “it has been reported that these effects are different based on the disease characteristics”

Line 155: Remove dot after table 2. Response: it has been removed “Table 2 Gastrointestinal Effect of Curcumin”

Lines 182-186: The type of letter is bigger than in the remaining document. Response:The font of these part has been adjusted

Line 189: Nelumbo nucifera should be in italic. Response:Nelumbo nucifera

Line 190: Curcuma longa should be in italic, remove the L since it was not added before. In addition, after mentioning the first time the authors should write “C. longa” in italic. Response: It has been corrected in all parts of manuscript. “C. Longa

Line 190: Replace provide for provided. Response:It has been replaced “provided”

Line 195: Add of after augmentation. Response: It has been added. “augmentation of the conversion”

Table 3: Explain what Tetrahydro-curcumin is.

Some of the studies in the tables are very old. It would be important to find more recent studies (last 5 years) for each disease.

Response:

The reference containing “Tetrahydro-curcumin” was old and removed. Some of old studies that were not necessary have been removed. 

 Line 207/210: use anti-cancer or anticancer, not both. Response:In all manuscript anti-cancer has been used

Line 222: Replace reveals for reveal. Response: It has been corrected. “The findings reveal”

Line 225: Remove a before low levels. Response: It has been removed. “showed low levels of curcumin”

Line 234: Define EGFR. Response:It has been defined “anti-EGFR (Epidermal growth factor receptor)”

Line 234: An or In a clinical trial? Response: This sentence has been rewritten “Furthermore, a combination of curcumin with anti-EGFR (Epidermal growth factor receptor) monoclonal antibodies in cutaneous squamous cell carcinoma patients has been showed as a highly effective strategy in disease control in another clinical”

Line 236: Patients a? Response: This sentence has been rewritten “Furthermore, a combination of curcumin with anti-EGFR (Epidermal growth factor receptor) monoclonal antibodies in cutaneous squamous cell carcinoma patients has been showed as a highly effective strategy in disease control in another clinical”

Line 246 (364, 419): Replace rout for route. Response:It has been replaced “In topical route of administration”

Line 261: Add the word disease. Response: Disease has been added “Alzheimer’s disease (AD) is the most”

Line 281: Define SUMO and JNK Response: they have been defined “by modulating of Small Ubiquitin-like Modifier (SUMO)-1-JNK(c-Jun N-terminal kinases)-TAU axis”

Table 6: explain what Curcumin C3 complex is and specify curcumin concentration and other curcuminoids.

Tables in general: It is only referred CT. Are the authors referring only clinical trials phase 1? Or is there phase 2? It should be specified.

Response: Curcumin C3 Complex(®) is a specific formulation; an extract derived from the rhizomes (roots) of the plant Curcuma longa, which has been defined in table

CT has been replaced by clinical trial. All these reports are phase 1.

Line 296: Replace this for these. Response: It has been corrected “for these effects”

Figure 2: The figure is a bit blurred. The figure quality should be improved. All abbreviations should be defined in the figure caption (VCAM, TNF, EGF….). Response: High quality version of figure has been uploaded.  All abbreviations have been defined in the figure caption.

Lines 301-302: Please revise this sentence. Response: It has been revised “The cellular and molecular targets of curcumin have been summarized in figure 2[190–193]. It has been categorized based on the role of targets and effect of curcumin on those.”

Line 350: Remove and. Response: It has been removed “for short periods of time so far”

Line 362: Remove the. Response:It has been removed “Whether this effect belongs to curcumin molecule”

Line 365/366: in m2 the 2 should be elevated (correct every time is used). Response: It has been corrected “a dose of 120 mg/m2

Line 372: Major should be in lower case. Replace curcuma longa for C. longa (in italic). Response:Both of them have been corrected “The major constituent of extracts of C. longa

Line 372-375: This is a repetition. It is already present in the beginning of the paper (lines 47-49). Response: This paragraph has been rewritten “As described previously, the major constituent of extracts of C. longa are called curcuminoids, which includes curcumin demethoxycurcumin, and bisdemethoxycurcumin, along with numerous and less abundant secondary metabolites[222].”

Line 384: Remove a. Response:it has been removed “there were just negligible quantities detected”

Line 390: Remove are. Response: It has been removed “Data shown in table 8 summarize”

Line 416: Define NADPH. Response: It has been defined “Nicotinamide adenine dinucleotide phosphate (NADPH)-dependent”

Line 417: Replace Escherichia coli for E. coli (in italic). Response:E. coli

Line 437-438: Revise sentence. Response: It has been rewritten “There are also some micronized formulations for curcumin, which were reported to have 9-fold increased bioavailable than unformulated.”

Line 439: Add which before increases. Response: It has been added “micronized curcumin has a smaller diameter of drug particles which increases the surface”

Table 8: This table needs a formatting revision (e.g. lower and upper cases). What do you mean with no in time maximum concentration? Why ref 249 has two different times? What is the unit of 33.4? Why using nM and in the other cases nmol/L? In ref 97 and 252, it is 3.6 or 3600 or 3 and 600? In ref 256 the authors presented 10,80,400 (are these 3 doses?). In ref 209 they wrote 1200, not 1,200…In ref 212 replace liter per L. This is confusing. Ref 257 and 261, g of what? Tissue? Ref 245, it should be mL instead of ml (correct other cases in the manuscript). Last reference from page 21 is not ok

Response: All corrections have been done: 1. format revision for lower and upper cases. 2. no has been replaced by No data 3. Ref 249 has been checked   4. Reference 249 has been corrected 5. The unit for “33.4” has been corrected. 6. All nmol/L have been changed to nM 7. They have been corrected 3600 mg and three doses 10, 80, 400 mg. 8. Liter has been replaced to L. 10. 257 and 261, g of tissue which has been added. 11. Ref 245, ml has been replaced to mL. in other parts of manuscript also corrected. 12. Last reference from page 21 has been revised.

Line 475: Replace has for were. Response:It has been replaced “According to the therapeutic effects that were summarized”

Lines 500-504: Please revise the paragraph and pay special attention to expressions such as “need to be studied more”, “limit number of investigates” and “in other organs” (what were the other organs?). Response: the paragraph revised and has been changed “Although the absorption and bioavailability are the crucial issues, curcumin distribution in body tissues should be take into account for its biological activity. In fact, there are just limited number of investigations have addressed this problem so far. In this regard, researchers reported that oral administration of curcumin (400 mg) in rats exerts only traces of unchanged drug in other organs such as liver and kidney [223].”

Line 509: Revise sentence. Response: This sentence has been rewritten “The data on pharmacokinetic parameters and clinical studies reported in this review, revealed that better absorption of curcumin and related formulations is highly depends on improving the solubility and decreasing the rapid first-pass metabolism.”

Line 511: Population of objects?? Response: It has been replaced by “sample size”

Line 525: Add author contributions. Response:author contributions section has been added to the manuscript.

References: The manuscript has 261 references and several are quite dated. Primary research from the past 5 years should be prioritized. Response: all references have been revised some old ones have been removed. For some effects there are not more recent (past 5 years) studies so we had to use some old references such as anti microbial effects part.

Round 2

Reviewer 1 Report

I have no further comments. All my questions have been addressed.

Author Response

Thank you very much

Reviewer 3 Report

Dear Authors,

Now I understand what authors tried to describe.

I think the review article could be useful for increasing and updating our knowledge on curcumin problems and try to overcome its limitations.

Thank you.

Author Response

Thank you very much

Reviewer 4 Report

The authors did not mark in yellow all the corrections made. At least small corrections were not marked.

In several places in the manuscript appears rout instead of route or routes.

Line 41 – Remove C. longa. The nomenclature is not as a normal abbreviation. The first time it appears in the introduction should be Curcuma longa and afterwards C. longa (always in italic). It does not need the information in the brackets.

Comment related to line 49: “It means that among the 3 main curcuminoids, curcumin (75–80%), which is the main colored compound and the yellow color of this compound depends on the presence of curcumin.“ - This is not correct, all three main curcuminoids are yellow therefore the sentence should be corrected, including the modification the authors made next.

Figure 1 caption – replace disease for diseases

Line 81: Replace “effect” for effects

Line 83: Replace “reports” for reported.

The title “Disease targets of curcumin: from cell lines to clinical trial studies” letter size/type should be different for the next subsections (e.g. Antimicrobial Effect of Curcumin).

Line 102: Akt meaning is still missing. Sometimes the authors write Akt and others AKT. They should opt for one version.

Table 1: Aspergillus parasiticus, Candida albicans, Giardia lamblia and other organisms should always be in italic; HIV and HIV-1 is the same? If so, use the same way to refer to HIV.

These comments were not attendedLine 131: Remove dot after table 2”/ Line 155: Remove dot after table 2.

Table 2: in the first study inhibiting should be in lowercase. Where it says “Prevention of the Oxidative Stress Induced by Chronic Alcohol” – the sentence should be in lowercase.

Table 3: “co-administration” should be in lower case; In “Cardioprotective: significantly decreased MI associated with coronary artery bypass grafting through The antioxidant and anti-inflammatory effects” – “the” should be in lower case. There are mistakes like this in all the tables that should be corrected

Table 4: In the first study – “by maintaining” – by maintaining what?

Table 6: Curcuma longa should be in italic.

Table 7: There are words in Bold. This is not consistent with the other tables.

All tables should be reviewed in order to find formatting errors and inconsistencies. In addition, there are several abbreviations in the tables without the meaning.

Line 353: mail metabolite?

Line 367: It is not a paragraph

Line 551: Replace have for that.

Author Response

Dear Reviewer,

we really want to thank your for your precise MS revision. This has improved a lot out paper.

I reply below to your questions: 

R.The authors did not mark in yellow all the corrections made. At least small corrections were not marked.

A. Now we have marked all the corrections you suggested

R. In several places in the manuscript appears rout instead of route or routes.

A. Now we have corrected this point

R. Line 41 – Remove C. longa. The nomenclature is not as a normal abbreviation. The first time it appears in the introduction should be Curcuma longa and afterwards C. longa (always in italic). It does not need the information in the brackets.

A. Now we have corrected this point through all the MS

R. Comment related to line 49: “It means that among the 3 main curcuminoids, curcumin (75–80%), which is the main colored compound and the yellow color of this compound depends on the presence of curcumin.“ - This is not correct, all three main curcuminoids are yellow therefore the sentence should be corrected, including the modification the authors made next.

A. Now we have corrected this point 

R. Figure 1 caption – replace disease for diseases

A. This point has been corrected

R.Line 81: Replace “effect” for effects

A. This point has been corrected

R.Line 83: Replace “reports” for reported.

A. This point has been corrected

R.The title “Disease targets of curcumin: from cell lines to clinical trial studies” letter size/type should be different for the next subsections (e.g. Antimicrobial Effect of Curcumin).

A. This point has been corrected

R.Line 102: Akt meaning is still missing. Sometimes the authors write Akt and others AKT. They should opt for one version.

A. This point has been corrected

R.Table 1: Aspergillus parasiticus, Candida albicans, Giardia lamblia and other organisms should always be in italic; HIV and HIV-1 is the same? If so, use the same way to refer to HIV.

A. This point has been corrected

R.These comments were not attended “Line 131: Remove dot after table 2”/ Line 155: Remove dot after table 2.

A.This point has been corrected

R.Table 2: in the first study inhibiting should be in lowercase. Where it says “Prevention of the Oxidative Stress Induced by Chronic Alcohol” – the sentence should be in lowercase.

A.This point has been corrected

R.Table 3: “co-administration” should be in lower case; In “Cardioprotective: significantly decreased MI associated with coronary artery bypass grafting through The antioxidant and anti-inflammatory effects” – “the” should be in lower case. There are mistakes like this in all the tables that should be corrected

A.This point has been corrected

R.Table 4: In the first study – “by maintaining” – by maintaining what?

A.This point has been corrected

R.Table 6: Curcuma longa should be in italic.

A.This point has been corrected

R.Table 7: There are words in Bold. This is not consistent with the other tables.

A.This point has been corrected

R.All tables should be reviewed in order to find formatting errors and inconsistencies. In addition, there are several abbreviations in the tables without the meaning.

A. we have formatted them but we think to extend all abbrevations would make the table to long and in case a reader is interested can read the related cited papers in a more extense way

R.Line 353: mail metabolite?

R.Line 367: It is not a paragraph

R.Line 551: Replace have for that.

A. All the above corrections has been done